# How China’s Great Bay Area Policies Affect the National Identity of Hong Kong Youth—A Study of a Quasi-Natural Experiment Based on the Difference-in-Differences Model

**DOI:** 10.3390/bs13080644

**Published:** 2023-08-01

**Authors:** Chengzhe Fu, Liao Liao, Tingyang Mo, Xiaoqing Chen

**Affiliations:** 1School of Politics and Public Administration, South China Normal University, Guanzhou 510006, China; fcz_45@163.com; 2Division of Public Policy, Hong Kong University of Science and Technology, Hong Kong 999077, China; xchenep@connect.ust.hk

**Keywords:** political–cultural adaptation, national identity, China’s Bay Area policies, HK youth, DID model

## Abstract

China’s Bay Area policies are important for integrating Hong Kong (HK) and Macao youth into China’s overall development. However, their effect on national identity is still mostly theoretical, lacking objective and scientific empirical evaluations. From a cultural adaptation perspective, interactions between social groups with different cultural backgrounds can promote cultural and political identity transformation. Therefore, guided by China’s Bay Area policies, which encourage various cross-border exchanges via economic cooperation, HK youth can keep in touch with such Mainland Chinese cultural values as “responsible government”, facilitating “political socialization”, and political–cultural adaptation, helping to promote their psychological inclusion into society, contributing to a positive attitude towards the mainland government, and achieving the policy effect of building national identity. A quasi-natural experiment based on the regional differences in the first stages of China’s Bay Area policies can help to evaluate their effects on HK youth’s national identity. This study defines the initial stage of the Bay Area policy implementation as from 2016, when the 13th Five-Year Plan advocated building the “Great Bay Area of Guangdong, Hong Kong and Macao”, to 2019, when the Outline was published. The policies issued at this stage were the so-called “early policies”. Due to data limitations, it is difficult to obtain post-2019 survey data; therefore, the study mainly focuses on the policy effects at the beginning of the Bay Area’s construction. Four groups of cross-sectional data from the World Value Survey 6 (WVS6) (2010–2014) and World Value Survey 7 (WVS7) (2017–2020), with HK and the mainland (the non-Guangdong region) included, are used to examine the policy effect under the Difference-in-Differences (DID) Model. The research shows that the policies significantly enhanced HK youth’s national identity, and their trust in the mainland government played an intermediary role in the policy effect mechanism. However, the effect was uneven, benefiting the national identities of HK youth working in the for-profit private sector more than their unemployed or public sector peers. Therefore, this research proposes several policy implications to facilitate policy decision making related to youth in China’s Bay Area.

## 1. Introduction

Since Hong Kong’s return to the PRC in 1997, the CPC and the country have focused on constructing and promoting HK youth’s national identity in governing HK and Macao. The “*Outline Development Plan for the Guangdong–Hong Kong–Macao Greater Bay Area*” (hereinafter referred to as the *Outline*), which we call China’s Great Bay Area (GBAGBA), prioritizes working to “enable compatriots in HK and Macao to share with the people in the motherland both the historic responsibility of national rejuvenation and the pride of a strong and prosperous motherland” and “encourage young people to socialize, exchange, talk and mingle with one another, and support the integration of the youth of HK and Macao into the country and their participation in the country’s development”. In the academic field, there are also many studies proving the importance of cultivating national identity for personal development, group harmony, and the stable development of the national community. National identity, as an important expression of the legitimacy of the national community, has important positive significance. In terms of intergroup relations, groups can bring about the improvement of intergroup relations and the elimination of intergroup conflicts by reorganizing social classification and constructing a common identity [1]. Under the institutional background of expanding and deepening exchanges in the Bay Area, building a common identity can maintain stable intergroup relations between the mainland people and Hong Kong citizens and avoid the occurrence of intergroup conflicts and contradictions. In terms of personal development, empirical studies have proved that national identity encourages Hong Kong youth to participate in mainland exchanges and return to the mainland to seek personal career development opportunities [2] indicating that the enhancement of HK youths’ national identity through strengthening the support of people and deepening their national awareness has become an important development objective of the GBAGBA, and it is also an important guarantee for the stable and long-term implementation of the One Country, Two Systems principle.

The *Outline*, a guiding document for the top-level design of the GBA’s coordinated development, is an important symbol of the maturity of the GBA’s construction. However, long before its launch, GGBA policy implementation had already begun. In March 2016, the 13th Five-year Plan asserted that HK and Macao would be supported to play an important regional cooperation role in the Pan-Pearl River Delta and that platform construction for major cross-provincial cooperation in the GBAGBA would be promoted, marking the first time that the GBA concept had appeared in a Chinese government document. In 2017 and 2018, China’s State Council reiterated this, pointing out that the integration of HK and Macao into China’s national development should be supported by building the GBA (the content of the government reports is available at: http://www.gov.cn/guowuyuan/2017zfgzbg.htm and http://www.gov.cn/guowuyuan/2018zfgzbg.htm) (accessed on 1 May 2023). In 2017, the “*Framework Agreement on Deepening Guangdong–Hong Kong–Macao Cooperation in the Development of the Greater Bay Area*” was signed in Hong Kong, leading to a series of GBA-related policies and measures. Thus, the period from the implementation of the 13th Five-year Plan to the announcement of the *Outline* in 2019 can be considered the initial stage of the construction of the GBA, and the specific policies and measures issued in this stage can be called early policies. Since then, the central government, together with the People’s Government of Guangdong Province and the governments of the HK and Macao Special Administrative Regions (SARs), focusing on the theme of constructing the GBA, have introduced policy measures in such fields as education, innovation, and entrepreneurship and enhanced social services, providing abundant opportunities for young people and giving them a window of opportunity to get to know the mainland and build their national identity.

With the implementation of early policy measures, the result and mechanism of China’s Bay Area policy effect have gradually attracted attention from all walks of life. However, there is no definite conclusion or consensus regarding the policies’ effect on national identity and people’s support. Many scholars have drawn positive conclusions through their research. Chengzhe et al. found that the policies encourage HK and Macao youth to seek opportunities by going north to the mainland and promote their integration into the overall national development, which has integrated people’s support psychologically [3]. Minxia et al. pointed out that under the GBA cooperation framework, cultural exchanges between youth, varied in forms and rich in content, can promote cultural integration, enhance cultural awareness, and strengthen HK and Macao youth’s national identity [1]. Other scholars have put forward different opinions based on the status quo and development trend of HK youth’s identity. For example, Bixia found that policies oriented toward economic cooperation have not worked well because of HK society’s post-materialism and low population turnover rate [4]. The research team of Dongping et al. asserted that the 2019 Anti-Extradition Law Amendment Bill Movement (the Movement) reflected that the national identity predicament of HK youth is the result of an institutional identity predicament that hinders the construction of the GBA and the stable practice of the One Country, Two Systems principle [5].

Viewed from the perspective of China’s policy practice continuity, a series of GBA policies implemented after the *Outline* was issued are largely continuations of specific policies and measures introduced in the early stage, and the policy support is more powerful than before. Concerning the extreme inversion model put forward by King et al. in *Designing Social Inquiry: Scientific Inference in Qualitative Research* (that is, the method of extrapolating the overall situation through extreme cases [6]), the minimum policy effectiveness after the *Outline* was issued can be inferred using the evaluation results of the initial stage policy effects as a benchmark reference. Early-stage GBA construction policies involved few areas and were less specific; deep exchanges and cooperation in the GBA were not yet achieved. If it can be proved that the early-stage GBA policy positively affected HK youth’s national identity, the effect of the GBA policies issued since the *Outline*, which had an increased implementation area and strength, can also be inferred.

Therefore, from a Behavioral Public Administration (BPA) standpoint, the effective evaluation of the early policies (this study defines the period from the proposal of constructing the Guangdong–Hong Kong–Macao Greater Bay Area in the 13th Five-Year Plan to the release of the *Outline* in 2019 as the brewing stage for the implementation of the Greater Bay Area policies. The policies issued at this stage are referred to as early policies.) for GBA construction can provide a reference for and insight into the effective adjustment and improvement of the policy mechanism, which could have greater future effect. Therefore, from the perspectives of institution and psychology under BPA, this paper focuses on whether and how early GBA policies affected HK youth’s national identity. Specifically, it uses large-scale social survey data to construct a quasi-natural experiment framework that evaluates the policies’ behavioral effect on HK youth, reveals their underlying mechanism to find the right path to build HK youth’s national identity with different characteristics, and proposes targeted suggestions. It also explores public policy’s shaping effect (as an intervention) on political psychology from a cultural psychology perspective, examines its transformation mechanism’s effects on national identity from a cultural value changes perspective, enriches the BPA’s application paradigm with cultural psychology research, and engages in a dialogue with the existing national identity research to make contributions to the fields of national identity and behavioral public management.

The paper’s basic structure is as follows. The first section introduces the study, after which the second section briefly presents the theoretical basis to evaluate HK youth’s national identity and policy mechanism and puts forward the corresponding hypotheses. The third section introduces the DID model’s design and the variables’ descriptive statistics. The fourth section then shows the structure of the empirical analysis and justifies the research hypothesis, and finally, the fifth and final section presents a conclusion on the study.

## 2. Theoretical Basis and Research Hypothesis

### 2.1. National Identity from the Perspective of Acculturation

Acculturation was first studied in the early 20th century [7,8] by anthropology and sociology researchers who regarded it as a group phenomenon. For example, Redfield et al. believed that cross-cultural adaptation comprised changes in two cultural patterns due to successive contacts between groups composed of individuals from two cultures [9]. However, social psychologists regarded acculturation as an individual phenomenon. Berry classified acculturation into four categories (integration, separation, assimilation, and marginalization) based on the adapters’ attitudes towards their culture of origin and social membership in the new group. Integration occurs when individuals want to maintain their original cultural identity and cultural characteristics but also establish and maintain a good relationship with members of the host society and share their identity. Assimilation, conversely, refers to individuals attaching importance to communication with the dominant group and having a sense of belonging to the common national identity but abandoning their original identity [10]. From this perspective, establishing identity signifies individuals’ high-level acculturation and integration into new cultures and groups. On the other hand, Social Identity Theory holds that individuals establish identity through contact with internal and external groups in social communication, including national identity through social categorization, social comparison, and positive distinctiveness [11]; therefore, identity reveals the relationship between individuals, groups, and society [12], while national identity is formed by the interaction between individuals and groups in the national community and influenced by macro-institutional elements and micro-psychology at the same time. At the macro-social level, identity exists in the specific social environment and broader historical and cultural background [13]; at the micro-individual level, various factors of the macro-social environment provide important materials for the construction of individual national identity by shaping the collective memory [14] and affecting different societal members’ resource possession and development opportunities [15]. From the perspective of macro institutional and micro psychological interactions, the macro system policy guides the changes in micro individual behavior, which is the psychological root of effective social governance. On the issue of national identity, the macro Bay Area policy has changed the material and spiritual living conditions of Hong Kong youth to varying degrees in some fields, creating opportunities for them to contact with the mainland society and mainland compatriots, thus shaping their cognition of mainland society and the mainland government and building national identity. Based on this, it can be considered that national identity reflects the indicators that affect individuals and impact their psychology through macro-social changes and policy interventions, thus reflecting the mapping of macro-cultural changes to individual identity.

Most current acculturation studies focus on how people, including overseas students and immigrants, choose and adjust their behavior in a new environment after completing the socialization process in a cultural environment. However, along with the continuous advancement of the GBA’s coordinated development, increasing cross-border exchange activities and flows of people have caused conflicts between HK and mainland culture and increased the challenge of acculturation. According to studies, from a policy perspective, acculturating HK youth refers to improving their understanding of mainland culture, eliminating intergroup conflicts, and dispelling their stereotypes of their mainland compatriots by participating in exchange activities in the GBA and being exposed to policy promotion. In this process, changes happen in their cultural concepts, and they gradually identify with the mainland culture and then build a national identity. From the acculturation perspective, studies assert that national identity reflects the individual’s recognition of the country’s dominant culture, system, and main population and is an important indicator for evaluating their degree of acculturation.

### 2.2. The Effect of the GBA Policies on National Identity

Most traditional public administration studies regard cultural values as social background factors informing government behavior and policy formulation and implementation. The cultural effect and the values’ guiding function for public policy are relatively ignored. As the authoritative distribution of social value, public policy actively guides individual behavior and social practice by allocating social resources and creating public activity situations. It can imperceptibly promote a change in participants’ values in new practices and bring about cultural change and identity transformation.

National identity originates from people’s cognition and is influenced by factors such as mental activity, ideology, and belief. It refers to people’s acceptance of and obedience to the country in which they live [16]. Working from a psychological perspective, some scholars have found that national identity changes with individual experience and the environment. Changes happen from cognition to emotion and then to behavior [17]. National identity reflects an individual’s recognition of their country’s mainstream culture, system, and the main body of the population. It is an important indicator of the degree to which a person identifies as being a member psychologically.

The formation of HK youth’s national identity and the effect of policies thereon can be explained by the concepts of in-group and out-group and the theory of intergroup contact. According to the concepts of in-group and out-group, individuals compare and analyze their in-group and out-group when categorizing themselves; the more differences individuals perceive between their in-group and their out-group, the more readily they can perceive their similarities with the former. Thus, they build identification with the in-group by perceiving differences and similarities. In negative intergroup contact, threats from external groups can also enhance individuals’ identification with internal groups [18].

Over its long period of British colonial rule and political isolation from the mainland, HK developed different regional characteristics in its politics, legal system, economic system, social values, and ways of contact. The One Country, Two Systems principle practiced after the return has preserved the characteristics of HK society as much as possible, enabling HK youth to perceive the differences between the two sides, divide their in-group and out-group, and construct their identity. Qin et al. empirically analyzed the role of “others” (external groups) in HK youth’s national identity from a core–periphery perspective; since the 1970s, HK people have gradually developed a local consciousness in which they are convinced of the otherness of the mainland and their vast difference from Mainland Chinese people [19]. Due to the great differences in the societies and their group characteristics, HK youth tend to see HK people and mainlanders as internal and external groups, making finding more perceptual similarities difficult and hindering the establishment of a common identity.

However, intergroup contact theory holds that effective contact between different groups can improve mutual understanding, eliminate misunderstandings and stereotypes, build consensus and emotional bonds, and facilitate identity building. Empirical research has also proven that positive intergroup interactions such as community participation and interaction with local people can enhance the identification of immigrant groups with the country of origin [20]. From the psychological perspective of BPA, the policies’ effect on national identity is reflected in the meaningful integration in different dimensions of culture, politics, and interests that inform different types of identity integration, the mechanism of which can be built up through social and psychological efforts [21]. Integration in the interest dimension refers to the mutual benefits attained through economic exchanges and cooperation between the two sides, which is conducive to identity integration. Integration in the political dimension, the core of citizenship, mainly focuses on the integration of Western political ideology and Chinese political concepts. Integration in the cultural dimension, centering on people, finds expression in the integration of HK’s Western-oriented cultural and psychological values with those of the mainland, which are rooted in traditional Chinese culture. 

Regarding economic interests, under the GBA policies, which have broken down institutional barriers and accelerated interconnections between people, logistics, and capital flows, an international technology innovation center was built, and HK youth were encouraged to go north for entrepreneurship and employment. Thus, by constructing the GBA, their sense of gain and happiness can be enhanced and their national identity built. In the civic and political dimension, exchanges and cooperation in public administration and social governance have been deepened, and the path, theory, and system of socialism with Chinese characteristics have been comprehensively introduced through multiple channels to help HK youth correctly understand the differences between Western political concepts and the Chinese political system and enhance their understanding of and trust in the mainland government, which is conducive to forming and consolidating political identity. In the ethnic and cultural dimension, common cultural resources among regions and ethnic groups in Guangdong, Hong Kong, and Macau are used to organize educational programs, cultural and creative activities, and other cultural exchange events to enhance HK youth’s national pride and evoke their sharing national and cultural memory with their mainland compatriots.

Many field studies have proved that having HK youth work and study in Mainland China can effectively promote economic, political, and cultural integration, thereby enhancing their national identity. For example, some studies have shown that frequent visits to the mainland help improve their national identity, with psychological integration mediating this process [3]. In recent years, mainland governments at all levels have been orienting the GBA’s integrated development, resulting in bilateral population mobility, employment, and entrepreneurship, as well as cultural and academic exchanges that help HK youth recognize their shared identity with the mainland in such areas as cultural values and economic interests. HK youth’s national identity is enhanced by addressing institutional differences and cultural heterogeneity. Accordingly, Hypothesis 1 is proposed:

**H_1_:** 
*The GBA policies can significantly enhance HK youth’s national identity.*


### 2.3. The Role of Cultural Adaptation and Government Trust in Policy Effect

Babiker et al. (1990) proposed the concept of cultural distance, pointing out that the difference between the sojourners’ original culture and the local culture regulates their process of integrating into the new environment [22]. According to cultural distance theory, the greater the distance between a sojourner’s home and host culture, the more difficult their cross-cultural adaptation will be. Therefore, this study discusses the adaptation of HK youth to mainland culture under policy intervention from the perspective of changes in the cultural distance. Most existing research has explored the impact of cultural distance on enterprises’ overseas development [23] and transnational trade [24] from a macro point of view; few studies have analyzed the change in the cultural distance between individuals and new social environments and the causes thereof at the individual level. Administrative culture plays an important role in acculturation since it is public in nature. Administrative organs and staff behavior dominate social public life, unconsciously reflecting and shaping the unique local administrative culture.

At present, there are two main explanations for the source of government trust: institutional theory and cultural theory [25]. The former believes that government behaviors, performances, and policy interventions are the sources of public trust in the government. The latter believes that government trust is a special form of social trust influenced by long-standing social culture, value systems, and social capital. Combining the above two points of view, this study believes that policy intervention, against the background of the whole social culture, can achieve re-political socialization by encouraging the creation of positive conditions, thus changing individual cultural perceptions to a certain extent and building government trust.

Government trust refers to the people’s trust in and expectations of the government, gradually formed through long-term interactions. As an important part of political support, government trust is of great significance to national stability and political unity. The reproduction of political rituals is directly associated with an individual’s value orientation and cultural belonging and is an important carrier of national identity [26].

Hong Kong society and the mainland of China have experienced long-term political separation and opposition, forming distinct political cultures and ideologies due to their respective social systems. However, since HK’s return to China, the practice of One Country, Two Systems has guaranteed the SAR government a high degree of autonomy, objectively making it difficult for HK youth to have direct access to and form a correct fact-based understanding of the central government and the country’s main political system.

Xiaojun pointed out that the reasons for the political dilemma and identity crisis facing HK include the long-term poor communication and separation between HK and the mainland and the demonization of the mainland in HK’s political culture [27]. According to the theory of intergroup contact, the demonization can be ended by affording HK youth more contact with mainland governments through various exchange activities they organize and support. Only through effective contact can HK youth more clearly and comprehensively understand the aims and work styles of the mainland governments and their great achievements in social development (especially in the coordinated development of the GBA), realize the convenience and benefits attained by interacting with the mainland, and generate a sense of trust in the mainland government that lays the foundation for national identity.

The theory of administrative ecology holds that the administrative system and administrative behavior are in a specific situation, and a set of political and cultural symbol systems (including political myths, political norms, and political codes) determines the legal basis of the administrative system and provides a code of conduct for administrative agencies [28]. A society’s culture subtly influences the consciousness of administrative subjects (administrative organs and the staff) through various specific forms and content at all times, ensuring their administrative behavior follows and reflects society’s main culture values. Social culture and values that encompass administrative culture are the switches that determine the direction of an actor’s action [29] and provide judgment criteria for a series of behavior patterns [30]. If the sojourner can recognize the values behind local people’s behavior patterns, they will understand those patterns [31]. As the “outer group”, HK youth lack the experience of living on the mainland and need to actively learn the mainland’s social culture, especially its administrative culture, through participating in exchanges and acquiring contacts. Understanding and recognizing the mainland’s administrative culture can help HK youth understand the mainland government’s behaviors and build government trust based on a correct understanding.

This study used respondents’ self-rated variables in the WVS to assess the administrative cultural distance and the mainland government trust, that is, the variables the respondents filled in based on subjective perception. However, when subjective variables are used to explain subjective variables, the relationship between the two may appear spurious, due to confounding bias [32], possibly because the independent and dependent variables share underlying psychological traits [33]. Individuals’ unique psychological characteristics may simultaneously determine the value of the two subjective variables, thus causing confounding bias. To solve this problem, the study adopted two strategies proposed by Hu: theoretical debates and variable measurement. At present, cultural distance is measured using the values variable, while government trust is based on individuals’ basic attitude towards government agencies and staff. Values and attitudes represent different psychological tendencies with different sources and formation processes [34]. The former is shaped in an individual’s early socialization and belongs to a stable and subjective internalized culture [35]. The latter is more likely to be affected by short-term factors like government behavior and government performance in their personal public life. Thus, values are more stable psychological traits, while attitude are more variable psychological traits.

Many empirical studies believe national identity can enhance citizens’ political trust or trust in government institutions such as the police and their staff [36]. However, some scholars have come to the opposite conclusion. For example, by examining the changes in national identity in HK since the reunification, Steinhardt found that trust in the central government was the leading factor affecting national identity [37]. This study holds that government trust and national identity are attitude tendencies at two different levels. Political trust is a political emotional tendency based on political cognition, and citizens’ level of political trust is linked to realizing their interests, leading to instability in their behavior motivation [38]. As a special political trust for administrative organs, government trust is based on specific public life situations, such as government performance and administrative behavior, which are lower-level political attitudes. National identity is an individual’s sense of belonging to a certain group. Therefore, this study believes that government trust will continually affect individuals’ national identity.

As an important part of “political support”, government trust is of great significance to national stability and political unity. Reproducing the concept of political ceremony connects individuals’ value orientation and cultural attribution and is an important carrier of national identity [26]. As the country’s representative, the government evokes the common national cultural memory hidden in individuals’ hearts, plans and manages political and social life, constructs and reconstructs ideology, and finally constructs individuals’ national identity. People’s identification with a specific group, region, culture, and other aspects promotes the production of the political community. Under the One Country, Two Systems institutional framework, the mainland government can better represent China’s national image as an organizer of public life and a manager of public affairs (such as the coordinated development of the Bay Area) than the HK government. Therefore, a positive attitude towards the mainland government, the national image spokesperson, can improve national identity.

Political identity is the essential attribute of national identity, while cultural identity is the organic component of national identity. The two overlap. Given the identity of HK youth, some scholars have proposed that national education based on cultural symbols can enhance the cultural identity of HK youth and strengthen their national identity [39]. Therefore, the study believes that identifying national mainstream culture or shared values can help different groups explore perceived similarities in the process of social interaction, thus establishing national identity. In contrast, too much emphasis on the uniqueness of regional cultures and subcultures may weaken national identity.

To sum up, the Bay Area policies positively affect HK youth’s national identity building and deepen their understanding of mainland administrative culture through cross-border exchange activities and policy promotion, bringing them closer to it. By adapting to mainland administrative culture, HK youth would develop a correct understanding of the mainland government, building their trust in it and creating an identity like that in the national community. 

**H_2_:** 
*The Bay Area policies enhance HK youth’s adaptability to the mainland’s administrative culture, thereby increasing their trust in the mainland government, and ultimately national identity is built.*


### 2.4. Heterogeneity of the Policy Effect among Groups

Individuals in complex systems change their structure and behavior through repeated interaction with the environment; individuals’ adaptive behavior contributes to the system’s complexity [40]. People’s initiatives and organizations occupy an important position in social life, and adaptability leads to complexity [41].

The characteristics and initiatives of the subgroups in the policy target group lead to the heterogeneity of the policy effect. Policy cognition is the primary dimension of policy effect. Huaibin asserted that a policy’s behavioral effect needs to be explained from the perspective of cognition, recognition, and compliance [42]. The identity differences among public policy target groups, such as the knowledge level, age structure, education level, and occupation, lead to different cognition of the same public policy [43]. Policy target groups have varying opportunities to contact policy in their daily life, which leads to differences in policy cognition and the intergroup heterogeneity of the policy effects.

As mentioned above, the psychological mechanism of national identity is that social and psychological factors influence the degree of one’s national identity [21]. Against the background of the GBA’s coordinated development and the exchanges between the mainland and HK, HK youth interpret policies and their impact based on their personal identity and work and life experiences; this is how their national identity is built up. This paper holds that people’s employment status and the industry they work in are two identity factors that affect their cognition of macro policies at the psychological level. There are obvious differences in perceptions of policies on the three dimensions between the unemployed, those employed in the for-profit and non-profit private sectors, and those employed in the public sector, which lead to different degrees of national identity. The heterogeneity of policy cognition and acquisition will affect individuals’ sense of life gain in the fields of material life and social interaction, as employment status and employment departments provide living income for Hong Kong youth; while in the field of social interaction, they provide varying degrees of opportunities for youth to perceive and accept policy intervention. It has also been proven that job attributes can have a significant impact on personal happiness [44]. The latter will become the judgment standard for the construction of individual national identity [45]. In the brewing stage of the construction of the GBA, the exchanges between HK and the mainland were mainly themed around economic and trade cooperation, with national identity enhanced through the interests of economic development. For example, the main goals set in the “*Framework Agreement on Deepening Guangdong–Hong Kong–Macao Cooperation in the Development of the GBA*” signed by the National Development and Reform Commission and the governments of Guangdong, Hong Kong, and Macao in 2017 focused on such fields as science and technology, industrial innovation, finance, shipping, and trade. In this stage, national identity was enhanced by the policies’ focus on economic interests.

As for employment, young people who can participate in and benefit from economic cooperation in the Bay Area are often those in better economic conditions. The unemployed, however, did not enjoy the benefits brought by the policies. Many studies have taken HK youth working or studying in the mainland as the research objects to analyze the mechanisms influencing the national identity of this group. However, there is a sample selection bias in that young people who have not participated in or cannot participate in economic cooperation in the GBA have been ignored. Middle- and low-income groups, including the unemployed, cannot participate in the economic activities for the coordinated development of the GBA due to their limited economic conditions, abilities, and caliber and cannot build a national identity in such a situation. HK youth face deep-seated social issues such as sharp wealth differences, class immobility, and housing problems. and unemployed youth face greater living pressures. In response to the Movement, some studies also point out that slow income growth and increased living pressures have increased HK residents’ dissatisfaction with society, with some steering their dissatisfaction into real life and becoming the so-called mainstream opposition, influenced by the real opposition’s incitement and distortion. This has objectively led to more and more severe political turmoil [46]. Some political forces may induce unemployed youth to attribute HK’s social problems to the central government’s intervention and the differences and contradictions between the mainland and HK, thereby decreasing their national identity.

Regarding the industrial sector, young people employed in the for-profit private sector can enjoy more dividends from the early policies focusing on economic interest and the mainland’s development. However, the exchanges between HK and the mainland in the field of public services have not been significantly deepened and expanded at this stage. Young people employed in the public sector and non-profit private organizations, limited by the nature of their occupations, do not directly participate in economic activities nor can they directly benefit from the economic cooperation in the GBA. In contrast, in the civil and political dimensions, they noticed obvious differences in the systems of the two sides, which led to a decrease in national identity.

In addition, widening class differences and the growing gap between the rich and the poor have led to obvious heterogeneity among HK youth’s policy acquisition, exchange, and participation in the GBA. Studies in behavioral economics point out that the poor often make low-quality economic decisions because they lack cognition and the ability to make rational decisions [47]. Young people with low income may have difficulty crossing the policy threshold due to economic, cognitive, human capital, and social capital limitations, making it more difficult to leave HK to participate in the Greater Bay Area exchange and seek personal development opportunities. In contrast, attracted by the GBA policy, middle- and upper-class young people have more channels to participate in the GBA exchange by virtue of their relative social advantages.

Accordingly, Hypothesis 3 and the three sub-hypotheses are proposed:

**H_3_:** 
*The effect of the GBA policies on HK youth’s national identity is uneven.*


**H_3a_:** 
*The policy effect on employed youth regarding national identity is significantly higher than that on unemployed youth.*


**H_3b_:** 
*The policy effect is significantly higher on those employed in the for-profit private sector regarding national identity than those employed in the public sector and non-profit private organizations.*


**H_3c_:** 
*The GBA policy is more important in promoting the national identity of middle- and upper-class HK youth than the identity of those in the lower class.*


## 3. Research Design

### 3.1. Model Setting

In 2016, the 13th Five-Year Plan clearly stated that China would support HK and Macao in playing an important regional cooperation role in the Pan-Pearl River Delta and promote the construction of the GBA and major inter-provincial cooperation platforms. In 2017, the “*Framework Agreement on Deepening Guangdong–HK–Macao Cooperation in the Development of the GBA*” was signed in HK, and a series of policy measures to comprehensively promote the construction of the GBA was implemented. Accordingly, this study defines the period from the proposal of the 13th Five-Year Plan, where the concept of Guangdong–Hong Kong–Macao China’s Bay Area was put forward, to the release of the *Outline* in 2019 as the experimental stage of the implementation of the GBA policies. With HK youth as its research object, this study constructs a quasi-natural experimental framework and applies the DID model to test the net effect of the GBA policies on HK youth’s national identity.

The basic idea was to divide the survey samples into control and treatment groups. The object of the control group was those unaffected by the policies, and the object of the treatment group was those affected by the policies. The relevant data about the treatment and control groups before and after policy implementation were used to calculate the differences in certain indicators reflecting the policies’ effects. Then, the difference between the differences of the two groups, i.e., the double-difference estimator, was calculated, and the policy effect was evaluated accordingly.

The DID model requires constructing two sets of important variables, dividing the objects into two sets according to the coverage range and the time series. Regarding objects, this study explored the mechanism of the GBA policies enhancing national identity among youth. Therefore, respondents aged between 18 and 40 were selected as the study sample. The object of the treatment group was HK youth, and the sample code was 1. Since Guangdong and HK were both affected by the policies, the respondents in Guangdong were excluded, and youths from 30 other provincial-level administrative regions in the mainland were the object of the control group, coded as 0. Regarding time series, Wave 6 WVS respondents (2010–2014, before the GBA concept was proposed) were coded as 0; Wave 7 WVS respondents (2017–2019, i.e., after the GBA concept was proposed) were coded as 1. Thus, a mechanism model of policy effect was constructed that included the mediating variable, mainland government trust.
Yi = α + βTreated_i_ + γTime_j_ + δ (Time_j_ × Treated_i_) + μ_1_configovernment_ij_ + μ_2_X_ij_ + ε_ij_

Treated_i_ represents all groups, whether covered by the policies or not; i = 1 is the treatment group (HK youth), i = 0 is the control group (mainland youth except those in the Guangdong Province). Time represents the time; Time_j_ × Treat_i_ represents the interaction term of the dummy variables of time and policy, configovernment is the mainland government trust, and X is a set of control variables that change over time and may affect the mainland government and national identity.

The national identity of the control group is α + μ_1_ + μ_2_ before the policies’ implementation and α + γ + μ_1_ + μ_2_ after; the difference before and after the implementation of the policies is γ. The national identity of the treatment group is α + β + μ_2_ before the implementation of the policies and α + β + γ + δ + μ_1_ + μ_2_ after; the difference is γ + δ. The difference between the differences in the two groups is δ, the DID estimator, which represents the GBA policies’ effect on national identity. If δ is significantly greater than 0, the policies have significantly enhanced HK youth’s national identity.

### 3.2. Data Sources and Operated Variables

The data used in this study were from the Wave 6 and Wave 7 WVS, initiated by Ronald Inglehart, a professor of Political Science at the University of Michigan, and organized and managed by the World Values Survey Association (WVSA). The data were mixed from four groups of cross-sectional data in two periods in the mainland and HK. The WVS6 ran from 2010 to 2014 and the WVS7 from 2017 to 2019—just before and after the early policy measures for constructing the GBA, thus meeting the DID model’s time requirements. The WVS used probability sampling for large samples to reduce the differences caused by the different respondents in each period.

In this study, the dependent variable was national identity, and respondents were asked how close they were to China (very close = 1, close = 2, not close = 3, distant = 4). The study recoded this variable based on conventional logic and reversed the assignment of each degree of closeness (distant = 1, not close = 2, close = 3, very close = 4). The mediating variable in this study was mainland government trust. The specific object in the mainland questionnaire was administrative organizations (the mainland government, by default). The sample in the treatment group was directly asked questions reflecting their mainland government trust; the specific evaluation criteria were a lot of trust = 1, trust = 2, distrust = 3, a lot of distrust = 4. Reverse coding was performed as described above. The higher the two variables, the stronger the respondents’ national identity and mainland government trust.

The control variables in this study included sex (female = 0, male = 1), age, perception about personal income, feeling of freedom, and life satisfaction. The last three variables, as interval variables, were coded from 1 to 10 (low to high). The control variables also contained self-assessed social class and physical health. The two variables, as ordinal variables, were coded from 1 to 5 (low to high). The other control variables were employment status (employed = 1, unemployed = 0), marital status (with a spouse (married, cohabiting) = 1, without a spouse (single, divorced, widowed) = 0), education level, and post-materialist values. They were coded from 0 to 5 (low to high) as interval variables.

### 3.3. Descriptive Statistics

Table 1 shows the descriptive statistics of the variables used in this study, with the control and treatment groups divided according to whether the policy had been implemented. Among them, the demographic variables basically conformed to the population distribution characteristics of young people in the mainland and HK, indicating that the data reflected the characteristics of the population, and the parameters could be estimated based on the sample statistics. The proportion of the two categorical variables in Table 1 refers to the sample proportion with the variable being 0.

## 4. Empirical Analysis Results

### 4.1. Analysis Results Based on the DID Model

The study used the DID model to estimate the GBA policies’ promotion effect on the youth’s national identity with and without the control variables. The DID regression results are shown in Table 2. First, Model 1, which did not contain demographic and other control variables, only the effects of the Bay Area policy, was built as the benchmark model. Then, based on Model 1, the variables that might affect national identity, such as the personal characteristics variables, were included as Model 2. Finally, based on Model 2, the dependent variables were changed to administrative cultural adaptation and mainland government trust, and the impact of implementing the Bay Area policy on the two was preliminarily investigated. The statistical results were as follows.

It can be seen from the above table that the DID regression coefficients of the interaction items representing the policy effects of the GBA in Model 1 and Model 2 were positive, with the significance reaching the level of 0.01, indicating that the Bay Area policies introduced from the sixth round of survey to the seventh significantly positively impacted young people’s national identity. The DID interaction coefficient represents the average treatment effect (ATT) on the young people affected by the Bay Area policy. The ATT in the complete Model 2 was 0.141, indicating HK youth’s national identity groups increased significantly compared with youth from 30 mainland provinces and regions unaffected by the Bay Area policy, indicating that the Bay Area policy significantly improved HK youth’s national identity and verifying H1. The interaction coefficient representing the policy effects in Model 2 was relatively small, only 0.141, which may be caused by the following reasons. Firstly, the dependent variable national identity was a variable with a value range of 1–4, which may therefore reduce the overall change range of the variable. Secondly, it mainly examined the policy effects generated in the early stage of the Bay Area policy implementation from 2016 to 2019. Due to the short implementation time, the effect was relatively small. Finally, as a relatively stable psychological trait, national identity is difficult to strongly change in the short term. Therefore, short-term policy intervention can only have a relatively limited improvement effect on the cultivation of national identity in Hong Kong youth. In addition, after including the control variables in the model, the DID interaction term coefficient and R2 increased, indicating the demographic variables had some explanatory power for changes in national identity. Including these variables improved the model’s overall explanatory power.

To further explore the relationship between policy implementation, administrative cultural adaptation, and mainland government trust, the study conducted DID analysis on two intermediary variables. The results showed that in Model 3, with administrative cultural distance as the dependent variable, the interaction item coefficient was significantly negative, reaching a significance level of 0.01; in Model 4, which took mainland government trust as the dependent variable, the interaction term coefficient was significantly positive, reaching a significance level of 0.01, indicating that the Bay Area policy could significantly shorten the distance between HK youth and the mainland’s administrative culture and at the same time enhance mainland government trust, thus preliminarily verifying H2.

### 4.2. Robustness Test: Propensity Score Matching (PSM)-DID Model Results

The core hypothesis of the DID method is the parallel trend hypothesis, which requires a parallel trend assumption for the treatment and control groups; the explained variables are parallel or follow the same trend over time before the policy is implemented. In other words, if the policy does not affect the treatment group, the time trend is parallel to or the same as the control group. In such a condition, the researcher can approximate the control group as the counterfactual situation for the treatment group unaffected by the policies. Since the World Values Survey Association has conducted only two surveys in HK, there are only data from one period before and after the policy implementation, making it difficult to validate the parallel trend hypothesis. Additionally, there may be selection bias and systematic differences in demographic variables or other key variables between the control and treatment groups. There are obvious differences between the mainland and HK in terms of economic and social development and cultural values, which may interfere with the policy effect mechanism in the empirical results. Therefore, this study refers to the practice of Ahlfeldt (2018) [48] and Ke (2022) [49] in dealing with the parallel trend of two periods of DID, used the PSM method to strengthen the match between the control and treatment groups to avoid selection bias, and used both the DID and PSM methods (PSM-DID) to prove the GBA policies’ effect on youth national identity and the robustness of the mainland government trust’s mediation effect.

#### 4.2.1. Propensity Score Matching

The basic steps of the PSM-DID are as follows. First, the treatment variable D is taken as the dependent variable and the covariate X as the independent variable, using logistic or probit regression to calculate the propensity score P(x) of each subject, determine the region of common support, and detect the difference between the control and treatment groups after matching. This step reduces the dimensionality of multiple covariates that may affect the process variables and reduce the matching difficulty. Second, the two groups of samples are matched through nearest neighbor matching, caliper matching, kernel matching, and Mahalanobis distance matching (MDM). According to the results generated by different matching methods, the unmatched samples outside the common support region are deleted. Finally, the DID model includes the matched samples to test the policy effect mechanism.

To eliminate selective deviation, the study used the control variables as covariates and national identity as the dependent variable; the propensity score for each sample was calculated using the logit model. The results showed that the chi square statistic of the two matched models before and after policy implementation were 450.90 and 876.57.33, respectively, which were significantly different from zero, indicating the two matched models were effective (compared to the null model with no variables), with the pseudo R2 being 0.297 and 0.359, respectively, proving the model had a certain explanatory power.

#### 4.2.2. Balance Diagnostics after Matching

Based on the propensity score calculated by the above logit model, balance diagnostics were carried out on the treatment and control groups. The results in Table 3 showed that, except for sex and feeling of freedom, the difference between the matching covariates in the two groups decreased significantly, with a percent bias ranging from −60 to 90 before matching and tending to 0 after matching. The difference between the two groups passed the *t*-test (the *p*-value was larger than 0.10 for both), indicating that the matching result was good. Figure 1 and Figure 2 show that after matching, the standardized deviations of each matching variable in the two groups before and after policy implementation decreased, except for the feeling of freedom, the health condition, and post-materialist values. It can be assumed that there was no significant difference in most of the control variables between the treatment and control groups after matching, indicating the PSM-DID method was appropriate. At the same time, most of the samples were in the region of common support, meaning most could be matched, and there were few missing data in the propensity matching.

#### 4.2.3. Average Treatment Effect

There are many propensity score matching methods, including nearest neighbor matching, caliper matching, kernel matching, and Mahalanobis distance matching (MDM). This study used different matching methods to match the samples and estimate participation to ensure matching robustness. First, the results in Table 4 show that based on the results from the different matching methods, the samples within the region of common support were retained to determine the difference in national identity between the treatment and control groups.

The study used different matching methods. Nearest neighbor matching was used for Groups 1 to 3; the matching radius for caliper matching was 0.01. Before matching, the mean difference in national identity between the control and treatment groups was 0.256, with confidence reaching the 0.01 level, indicating the mainland youth’s national identity was significantly higher than the HK youth’s, in line with common sense. Except for the MDM, all other matching methods reduced or maintained the difference in national identity (between 0.161 and 0.352), while the T-value decreased. However, no matter how they were matched, there were significant differences in national identity between the control and treatment groups. After matching, there was still a large but decreasing difference in national identity between the two groups, indicating selection bias in the samples before matching. The PSM effectively eliminated the interference of the selection bias with the GBA policy effect mechanism.

#### 4.2.4. Robustness Test of the Policy Effect

This paper used the PSM-DID model to avoid selection bias and prevent different distributions of demographic variables in verifying the robustness of the Bay Area policy’s effect on national identity based on the sample after matching the propensity score. The results are shown in Table 5. Compared with Model 5, the R2 of Model 6 increased, indicating that including the control variables improved the models’ explanatory power, and the DID interaction terms coefficient increased from 0.191 to 0.202. Compared with the results before the propensity score matching, the Bay Area policy’s effect on national identity was somewhat framed after matching but still reached a significance level of 0.01. The results from Model 8 showed that the interaction terms’ coefficient reached a significance level of 0.01. The policy effect direction was consistent with the results before matching, conforming to the research hypothesis and indicating that the Bay Area’s policies had robustness on various effects of youth national identity and mainland government trust, after controlling for selective error, proving the robustness of H1.

### 4.3. Analysis of the Mechanism of the Policy Effects

The above verified the effect of implementing the Bay Area policies on promoting HK youth’s national identity. However, this study considered that there may not be a direct causal relationship between the independent and dependent variables but an indirect mediating variable. Therefore, structural equation modeling (SEM) was adopted to analyze further the relationship between the Bay Area policies’ implementation and HK youth’s administrative acculturation, mainland government trust, and national identity. Before using SEM to estimate the intermediary effect, it was necessary to test for model fit to ensure the model’s and the data’s suitability.

The results in the Table 6 show that the correlation between national identity, government trust, and mainland administrative culture has reached a significant level.

The results in the Table 7 show that the model fitting has reached the ideal level (χ2/df = 4.761, RMSEA = 0.035, CFI = 0.998, SRMR = 0.008, TLI = 0.982).The path coefficients in the Figure 3 showed that the GBA policy could shorten the administrative cultural distance between HK and mainland youth (β = −0.140, SE = 0.033, *p* < 0.01) and significantly enhance mainland government trust (β = 0.074, SE = 0.025, *p* < 0.01) and national identity (β = 0.110, SE = 0.032, *p* < 0.01). The narrow administrative cultural distance could significantly enhance youth’s mainland government trust (β = −0.027, SE = 0.013, *p* < 0.05), which could, in turn, enhance their sense of national identity (β = 0.209, SE = 0.025, *p* < 0.01). The Bay Area policies’ direct and indirect effects on national identity were verified. In addition to the direct path, Bay Area policies could enhance national identity through Bay Area Policy–Administrative Cultural Distance–Mainland Government Trust–National Identity and Bay Area Policy–Mainland Government Trust–National Identity.

### 4.4. Robustness Testing of the Subjective Variables Explaining the Subjective Variables

In analyzing the abovementioned policy effect mechanism, the relationships between the two subjective variables of administrative cultural distance and mainland government trust and between the two subjective variables of mainland government trust and national identity were discussed. Therefore, the study used Hu’s (2019) variable measurement strategy for robustness testing. First, the psychological trait of trust tendency might simultaneously affect individuals’ government trust and perception of government responsibility. Putnam (1993) pointed out that individuals’ general level of trust will affect their level of trust in government as a special subject. Individuals with low levels of general trust are more likely to believe that individuals should take more responsibility for their own lives rather than handing over responsibility to the government. Therefore, the study conducted factor analysis on WVS respondents’ trust levels towards their family members, neighbors, and other subjects, extracted the trust tendency factor, and added it to the analysis of three subjective variables to control the influence of the latent trust tendency on the analysis. The results are shown in Table 8.

Models 9 and 11 tried regressions with the mainland government trust and national identity, and Models 10 and 12 added trust tendency, based on Models 9 and 11, respectively. The comparison results showed that in Model 9, with mainland government trust as the dependent variable, the regression coefficient for administrative cultural adaptation was −0.036. After adding the trust tendency in Model 10, the regression coefficient became −0.030; the significance also decreased but maintained a 0.05 level. In Model 11, with national identity as the dependent variable, the regression coefficient of administrative acculturation was 0.185. After adding the trust tendency in Model 12, the regression coefficient dropped to 0.171, but the significance level remained 0.01. In addition, the coefficients of the interaction items in the four models were all significantly positive, reaching a significance level of 0.01. The absolute value of the regression coefficient of subjective variables declined but was still significant, indicating that administrative cultural distance’s influence on mainland government trust and mainland government trust’s influence on national identity were partly due to individuals’ psychological trust tendencies. However, after controlling for the trust tendency, there was still a significant correlation between the two pairs of subjective variables.

In addition, comparing model 10 with government trust as the dependent variable in Table 8 and the model with national identity as the dependent variable in Table 2, it can be found that the interaction coefficient in the government trust model was significantly higher than the corresponding coefficient in the national identity model, possibly due to their different levels of psychological attitudes. Government trust can be seen as the attitude of citizens towards specific government institutions, while identity belongs to a deeper field of values. Some scholars believe that values and norms are acquired through early socialization, but attitudes are formed by the interaction between values and norms acquired in the early life and the behavior of political and social actors (Shi, 2001). Since attitudes are partly affected by external stimulus, the change in the opportunity structure brought about by institutional and policy changes has a more obvious impact on attitude variables such as government trust.

### 4.5. Heterogeneity of the Policy Effect among Groups

To further test Hypothesis 3, the youth sample was divided into employed and unemployed based on the employment status variable, and DID tests were carried out. Table 9 shows that the interaction term coefficient was insignificant in the unemployed group (*p* > 0.10), indicating that the policies did not significantly improve national identity. However, it did significantly positively affect the employed group at a significance level of 0.01, indicating the GBA policies in the brewing stage significantly positively affected employed HK youth’s national identity but did not significantly affect HK’s unemployed youth’s identity, proving H3_a_.

In this paper, the employed youth sample, with the employment sector as the variable, was divided based on the work sector (i.e., public sector, non-governmental organizations (NGOs), and the for-profit private sector). The DID test was carried out to verify H3_b_. It can be seen from Table 10 that the interaction term coefficient was not significant in the public sector and NGO groups (*p* > 0.10), indicating the GBA policies did not significantly improve national identity, but they significantly positively affected the employed population at a significance level of 0.01, indicating that the GBA policies in the brewing stage significantly positively affected the national identity of the HK youth employed in the for-profit private sector but had no significant effect on those working in the public sector and NGOs. H3_b_ was proved.

The youth sample was divided into a lower-class group and a middle-and-upper-class group using the variable of social class, and DID tests were carried out on them, respectively. According to Table 11, the DID interaction terms’ standardized regression coefficients were significantly positive in both groups; however, the middle-and-upper-class group’s standardized regression coefficient was 0.137, reaching the significant level of 0.01, while the lower-class group’s was 0.116, reaching the significance level of 0.05. The interaction terms’ coefficient and DID significance level in the middle-and-upper-class group were higher than in the lower-class group, indicating that the early GBA policy had a promoting effect on HK youth from different classes’ national identity but a stronger effect on the middle-and-upper-class youths. H3c was proved.

## 5. Conclusions and Discussion

Based on data from the two World Values Survey waves in Mainland China and HK, this paper used the DID model to construct the mechanism of the effect on national identity with mainland government trust as the mediator variable. After controlling for demographic variables such as sex and age, it was found that in the early stage of implementing the GBA policies, HK youth’s national identity increased significantly, and mainland government trust mediated the policy effect mechanism. The effects were still robust after removing the endogenous effects through PSM. The DID test on the subsamples revealed the policy effect’s heterogeneity among groups. For instance, the effect on the employed was obvious, albeit more so for those employed in the for-profit private sector than in the public sector and NGOs. The above findings can be discussed and analyzed from the following three perspectives.

First, the GBA policies effectively achieved the goal of regaining people’s support. Many studies focus on the economic achievements made in the construction, level of infrastructure, and market connectivity of China’s Bay Area and explore the development patterns of city clusters and other economic development goals, but scant attention has been paid to the people’s integration at a psychological level due to economic cooperation and exchanges in the Bay Area. Research has found that the central government issued several policies to support HK youth seeking opportunities in the mainland and that increased cross-border exchanges have enhanced HK youth’s national identity [50], consistent with this paper’s findings. It can be assumed that preferential measures introduced in the initial GBA construction stage to encourage economic cooperation attracted some HK youth to actively participate in the country’s overall development, thereby effectively enhancing their national identity.

Next, building mainland government trust was important for enhancing the Bay Area policies’ effectiveness. Existing studies have defined cultural distance as the cultural gap between countries and regions [51] and analyzed its impact on international trade and cultural transmission. The mainland government is not only the country’s symbol but also the policies’ main implementer. Studies have pointed out that people’s national identity is often based on political trust in the country (government) and cognition of the traditional culture, social support system, and political legitimacy [42]. This study used DID modeling to prove the effect of government trust on national identity. Unlike previous studies, this paper also revealed the mechanism by which Bay Area policies enhanced national identity and explored the positive effects of shorter cultural distance at the individual level. The degree of acculturation was measured by the distance between HK youth and mainland administrative culture; it was found that administrative cultural distance and mainland government trust played partial mediating roles in the Bay Area policies’ national identity promotion mechanism. The mainland government instructs or guides the establishment of various integrated development platforms, such as the exchange centers for HK and Macau youth and innovation and business incubation bases in the nine mainland cities in the Bay Area, which is conducive to creating communication windows that allow HK youth to gain authentic information about the mainland government’s purpose, governance style, and achievements through various forms of exchange. In such an abundant social environment, they constantly come into contact with the mainland government’s unique administrative culture, promoting HK youth’s recognition thereof, helping to break their deep-rooted stereotypes about the mainland government, building their political trust in the mainland’s central and local governments, and enhancing their national identity (as represented by the central government). This finding shows that the existing GBA policies have borne remarkable fruit in guiding HK youth to understand the politics of the mainland and inspiring follow-up policies that continue to tell the GBA construction story to HK and Macao youths.

Finally, GBA policies must be personalized based on youth’s structural characteristics. In BPA, the policy behavior effect refers to the degree to which people’s behaviors change in response to public policies, based on their cognition and understanding of things [52]. In the acceptance process, policy objects passively understand and accept policies, building their cognition, recognition, and response to policies based on their life and work experiences and values. Therefore, to evaluate and analyze the policy effect and its mechanism, it is necessary to analyze the personal factors affecting the policy objects’ policy understanding and interpretation. Early GBA policies emphasized economic cooperation and mainly promoted HK youth’s integration at a psychological level through the economic interest dimension; however, HK youth are not a homogeneous group, and the policy effect was uneven due to sector and employment status differences. HK youth employed in the for-profit private sector had more opportunities to become involved in inter-regional economic cooperation and exchange activities guided by GBA policies, and their national identity was more likely to be enhanced by the policies. However, young people working in the public sector were less or indirectly involved in cross-border and economic cooperation in the Area, seldom enjoyed the dividends of national development, and were less affected by the policies. Having worked in HK for a long time, they tended to deepen and strengthen their identification with HK politics and justice, which reflect Western institutional heritage. Therefore, in the absence of public management exchanges in the Bay Area, this group may lack a correct understanding of the mainland system and often feel great institutional differences between the two sides, thereby losing their national identity. Regarding employment status, low-income youth, especially the unemployed, struggle to participate in Bay Area cooperation activities due to their limited abilities and resources and are thus less affected by the policies. Due to HK’s deep-seated social problems, this group faces great living pressures, and the resulting negative emotions translate into dissatisfaction with and hostility towards Mainland China and its central government because of the perceived differences and conflicts between the two sides, which hinder the building of national identity. Therefore, to improve policy effectiveness, personalized policies must be implemented based on the attributes of HK and Macao youth.

Based on the above analysis results and research conclusions, the main contributions of this study are as follows. Regarding subject dialogue, the study discussed the shaping effect of public policy as a political intervention from the perspective of political psychology and cultural psychology and analyzed the impact of policy intervention on national identity. Regarding functional mechanisms, most studies focus on macro factors like political system differences [53], education [54], and media discourse [55] to explore the causes of HK’s national identity crisis. Other studies move deep into the psychological level to find the logic underlying the construction of HK people’s national identity. Chengzhe et al. (2020) found that policy intervention promoted national identity through psychological integration [3]. Minxia et al. (2021) also proposed grasping the psychological mechanism of HK youth’s national identity by using psychological integration to eliminate identity estrangement [2]. In this study, the perspectives of political culture and value were introduced into the psychological mechanism of policy, and political and cultural psychological variables such as cultural distance and government trust were used to deeply analyze the political and psychological mechanisms of policy intervention in national identity, strengthening the explanation of this mechanism. Unlike previous studies, this study used two-period cross-sectional data from authoritative follow-up surveys; DID, PSM, and other methods were used to eliminate selective errors and make causal explanations for policy effects and their mechanisms more effectively.

Therefore, this paper provides some clues for policy makers. First, future GBA policies must continuously strengthen economic exchanges and cooperation between the mainland and HK and eliminate the institutional and psychological barriers to HK youth’s participation in the cooperation. Second, regarding public opinion, traditional mass media and social media should be used to show the mainland government’s true features and achievements in ways that HK compatriots will accept, so as to tell the story of the Bay Area’s construction and enhance HK and Macao youth’s trust in the mainland government. Last, the mechanism for building national identity should be constructed based on multiple dimensions and conditions. Those employed in the public sector should be encouraged to participate in China’s public affairs, with more activities being opened to deepen public administration exchanges between the two sides. For the low-income group, especially the unemployed, the mainland and HK governments should pay more attention to the deep conflicts in HK society, ensure people’s wellbeing, effectively solve problems related to people’s livelihood, and prevent the politicization of economic and social issues to avoid people’s negative emotions due to social problems, which hinder the building of national identity.

Of course, this paper also had certain limitations. First, only observed variables could be used in propensity score matching; however, unobserved factors might exist, leading to selection bias between the treatment and control groups. Second, due to data limitations, this paper only discussed the uneven policy effect from the two perspectives of employment status and employment sector. Additionally, it used data from two pre-2019 survey rounds and could not analyze the latest changes in HK youth’s national identity. Finally, traditional DID models use multi-period tracking data for causal inference [54], which involves collecting data from the same group of respondents before and after policy implementation. Due to limited data materials and a lack of better data sources, only two periods of non-tracking survey data could be used for analysis in this study. In the future, a timelier and more detailed database could be used to evaluate the policy effects and their mechanism in subsequent stages; in the analysis method, researchers can use a machine learning method such as random forest to include the massive variables in the analysis and explore the cultural and psychological mechanism of the policy effect on the individual country identity, to analyze the unique role of multicultural values in mediating mechanisms and to analyze the heterogeneity of policy effects among groups from different dimensions; in terms of data sources, we can collect the emotion, attitude, and behavior information reflecting the national identity of young people in Hong Kong in many ways, so as to provide multiple evaluation bases for the national identity cultivation policy and avoid the error of questionnaire measurement. It is imperative to analyze the heterogeneity of policy effect among groups from different dimensions to provide decision-making references for the personalized design of the GBA policies.

## Figures and Tables

**Figure 1 behavsci-13-00644-f001:**
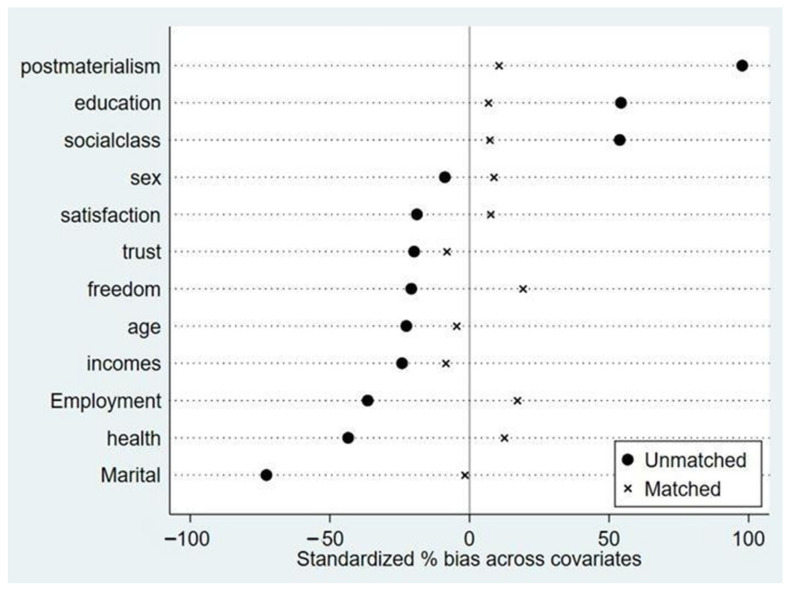
Standardized deviation of each matching variable before policy implementation.

**Figure 2 behavsci-13-00644-f002:**
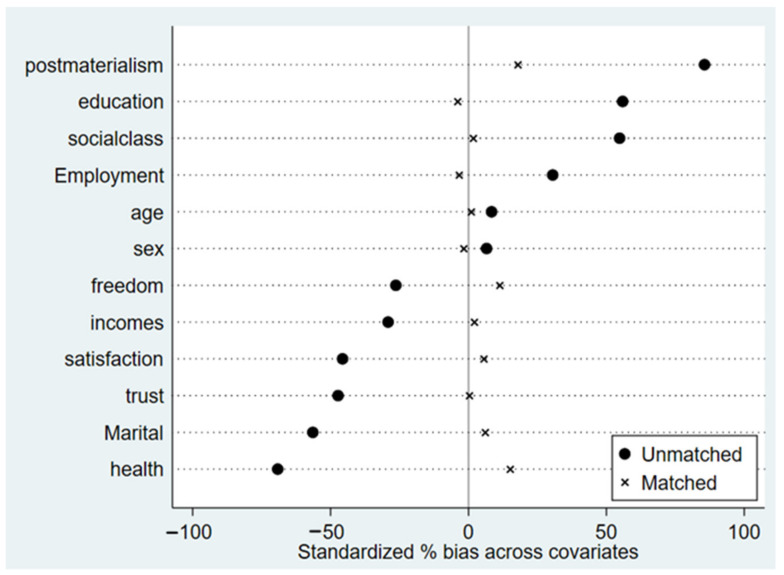
Standardized deviation of each matching variable after policy implementation.

**Figure 3 behavsci-13-00644-f003:**
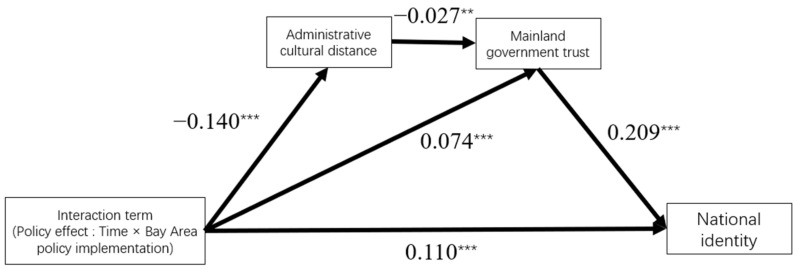
Path analysis diagram of the policy effect mechanism. Note: Dummy variables of time and policy implementation group were added to the model at the same time, which are omitted here to ensure a direct and concise path analysis diagram. In the figure, *** and ** respectively indicate that the correlation coefficients reached significance levels of 0.01 and 0.05, respectively.

**Table 1 behavsci-13-00644-t001:** Descriptive statistics (All empirical results in this article are based on the analysis of the youth sample).

	Entire Sample		Control Group: HK Youth		Treatment Group: Mainland Youth (Except Those in Guangdong Province)	
Variable	M	%/SD	M	%/SD	M	%/SD
Implementation of Bay Area Policy	0.252	0.434	1	0	0	0
Time	0.605	0.489	0.646	0.478	0.579	0.494
National identity	3.163	0.631	3.007	0.617	3.263	0.619
Mainland Government Trust	2.787	1.000	1.906	0.810	3.320	0.632
Administrative Cultural Distance	5.156	6.360	4.227	5.572	5.750	6.751
Sex	0.456	0.498	0.457	0.498	0.456	0.498
Employment Status	0.751	0.433	0.763	0.426	0.743	0.437
Marital Status	0.589	0.492	0.406	0.491	0.706	0.456
Age	29.933	6.740	29.780	6.851	30.031	6.668
Income	6.297	1.686	6.027	1.593	6.470	1.721
Social class	2.583	0.839	2.856	0.870	2.408	0.770
Education level	2.372	0.589	2.566	0.515	2.249	0.601
Self-assessed Feeling of Freedom	6.785	1.833	6.509	1.693	6.960	1.896
Health Condition	3.875	0.805	3.592	0.727	4.056	0.801
Life Satisfaction	6.837	1.853	6.440	1.665	7.090	1.922
Post-materialist Values	1.860	1.254	2.503	1.317	1.449	1.017

Note: M represents the mean value of the variable; % represents the proportion of samples with a binary variable value of 0 to the total sample; SD represents the standard deviation.

**Table 2 behavsci-13-00644-t002:** General linear model results.

Explained Variable	National Identity	National Identity	Administrative Cultural Distance	Government Trust
Variable	Model 1	Model 2	Model 3	Model 4
Time	−0.111 ***	−0.109 ***	0.230 ***	0.037 **
(Based on 2014)	(0.028)	(0.029)	(0.310)	(0.032)
Implementation of the Policies	−0.280 ***	−0.234 ***	0.029	−0.690 ***
(Mainland as the Control Group)	(0.037)	(0.040)	(0.432)	(0.050)
Interactive Term	0.127 ***	0.141 ***	−0.142 ***	0.089 ***
(Time × Implementation of the Policies)	(0.047)	(0.048)	(0.479)	(0.056)
Sex		0.022	0.046 **	0.023 *
		(0.023)	(0.231)	(0.026)
Age		0.038	−0.003	0.022
		(0.002)	(0.023)	(0.003)
Income		0.018	−0.024	−0.002
		(0.008)	(0.093)	(0.009)
Social Class		−0.014	−0.056 **	0.040 **
		(0.017)	(0.179)	(0.019)
Employment Status		−0.061 ***	−0.001	−0.042 ***
		(0.037)	(0.388)	(0.043)
Marital status		0.024	0.031	0.036 *
		(0.031)	(0.313)	(0.037)
Education Level		0.033 *	−0.088 ***	−0.040 ***
		(0.021)	(0.230)	(0.024)
Self-assessed Feeling of Freedom		0.013	0.042 *	0.007
		(0.008)	(0.080)	(0.009)
Health Condition		0.060 ***	0.033	0.026 *
		(0.016)	(0.160)	(0.019)
Social Trust		0.004	−0.042 **	0.071 ***
		(0.024)	(0.241)	(0.027)
Life Satisfaction		0.069 ***	0.037	0.057 ***
		(0.008)	(0.085)	(0.00901)
Post-materialist Values		−0.064 ***	−0.003	−0.107 ***
		(0.010)	(0.095)	(0.011)
Con	3.343 ***	2.852 ***	4.749 ***	2.923 ***
	(0.019)	(0.140)	(1.486)	(0.162)
R-sq	0.046	0.066	0.062	0.532
F	54.93	15.00	11.58	227.6
N	3032	2946	2946	2946

***, **, and * in the table indicate that the standardized regression coefficients reach the significance level of 0.01, 0.05 and 0.10, respectively. Standard errors are in parentheses.

**Table 3 behavsci-13-00644-t003:** Matching results of each variable.

	Percent Bias		
Variable	Before Matching	After Matching	Difference *t*-Test after Matching
Sex	6.6	−1.8	−0.32
Age	8.4	1.0	0.19
Income	−29.1	2.1	0.41
Social Class	54.8	1.7	0.34
Employment Status	30.5	−9.5	−0.70
Marital Status	−56.4	6.1	1.09
Education Level	55.9	−4.0	−0.80
Feeling of Freedom	−26.3	11.3	2.31 **
Health Condition	−69.2	15.1	2.82 ***
Social Trust	−47.3	0.3	0.05
Life Satisfaction	−45.7	5.6	1.05
Post-materialist Values	85.6	17.9	3.46 ***

**Table 4 behavsci-13-00644-t004:** Differences in national identity after matching.

Group	Matching Method	Sample	Control Group	Treatment Group	Difference	Standard Error	T Value
1	Unmatched	Unmatched	3.263	3.007	0.256	0.023	10.991 ***
2	1:1 nearest neighbor matching	ATT	3.205	3.044	0.161	0.041	3.962 ***
3	4:1 nearest neighbor matching	ATT	3.232	3.018	0.213	0.029	7.284 ***
4	Caliper matching	ATT	3.192	3.022	0.170	0.032	5.291 ***
5	Kernel matching	ATT	3.343	2.987	0.356	0.035	10.119 ***
6	MDM	ATT	3.204	3.021	0.182	0.031	5.908 ***

**Table 5 behavsci-13-00644-t005:** Results of the PSM-DID method.

Explained Variable	National Identity	National Identity	Administrative Cultural Distance	Government Trust
Variable	Model 5	Model 6	Model 7	Model 8
Time (Based on 2014)	−0.162 *** (0.055)	−0.162 *** (0.055)	0.187 *** (0.572)	0.021 (0.063)
Implementation of the Policies (Mainland as the Control Group)	−0.252 *** (0.048)	−0.243 *** (0.049)	−0.012 (0.541)	−0.676 *** (0.065)
Interactive Term (Time×Implementation of the Policies)	0.191 *** (0.068)	0.202 *** (0.067)	−0.112 * (0.678)	0.133 *** (0.079)
Cons	3.332 *** (0.036)	2.995 *** (0.201)	4.867 ** (2.036)	3.063 *** (0.255)
Control variable	Uncontrolled	Controlled	Controlled	Controlled
R2	0.031	0.047	0.047	0.450
F	20.90	6.055	4.587	103.9
N	1536	1536	1536	1536

***, ** and * in the table indicate that the standardized regression coefficients reach the significance level of 0.01 and 0.05, respectively. Standard errors are in parentheses.

**Table 6 behavsci-13-00644-t006:** Correlation coefficient matrix.

	1	2	3
1. National Identity	1.00		
2. Government Trust	0.240 ***	1.00	
3. Administrative Cultural Distance	0.041 **	0.064 ***	1.00

Note: In the table, *** and ** respectively indicate that the correlation coefficients reached significance levels of 0.01 and 0.05, respectively.

**Table 7 behavsci-13-00644-t007:** Standard values of the SEM goodness-of-fit index.

Goodness-of-Fit Index	Value	Evaluation Criterion
χ2/df	4.761	<5
RMSEA	0.035	<0.08
SRMR	0.008	<0.05
CFI	0.998	>0.9
TLI	0.982	>0.9
AIC	40,087.673	
BIC	40,189.962	

**Table 8 behavsci-13-00644-t008:** Robustness testing results of adding the latent psychological factor.

Explained Variable	Mainland Government Trust	Mainland Government Trust	National Identity	National Identity
Variable	Model 9	Model 10	Model 11	Model 12
Administrative Cultural Distance	−0.036 ***	−0.030 **		
	(0.002)	(0.002)		
Mainland Government Trust			0.185 ***	0.171 ***
			(0.018)	(0.018)
Time	0.046 ***	0.041 ***	−0.116 ***	−0.118 ***
(Taking the Year of 2014 as a Reference)	(0.032)	(0.032)	(0.028)	(0.028)
Bay Area Policy Implementation	−0.689 ***	−0.724 ***	−0.106 ***	−0.139 ***
(with the Mainland as the Reference Group)	(0.050)	(0.050)	(0.044)	(0.046)
Interaction Term	0.084 ***	0.087 ***	0.124 ***	0.125 ***
(Time × Bay Area Policy Implementation)	(0.056)	(0.056)	(0.047)	(0.047)
Trust Tendency		0.104 ***		0.071 ***
		(0.018)		(0.016)
Other Controlled Variables	Controlled	Controlled	Controlled	Controlled
_cons	2.950 ***	3.067 ***	2.510 ***	2.584 ***
	(0.162)	(0.160)	(0.152)	(0.153)
R-sq	0.534	0.543	0.082	0.087
F	214.3	212.3	16.37	16.10
N	2946	2938	2946	2938

*** and ** in the table indicate that the standardized regression coefficients reach the significance level of 0.01 and 0.05, respectively. Standard errors are in parentheses.

**Table 9 behavsci-13-00644-t009:** Classification of the youth samples by employment status.

Explained Variable	Employed	Unemployed
Variable	Model 13	Model 14
Time (Based on 2014)	−0.109 *** (0.031)	−0.154 *** (0.076)
Implementation of the Policies (Mainland as the Control Group)	−0.247 *** (0.044)	−0.203 *** (0.094)
Interactive Term (Time × Implementation of the Policies)	0.143 *** (0.053)	0.154 * (0.113)
Cons	2.644 *** (0.156)	3.547 *** (0.379)
Control variable	Controlled	Controlled
R2	0.071	0.083
F	14.16 ***	3.870 ***
N	2482	464

***, **, and * in the table indicate that the standardized regression coefficients reach the significance level of 0.01, 0.05 and 0.10, respectively. Standard errors are in parentheses.

**Table 10 behavsci-13-00644-t010:** Classification of the employed population by sector.

Explained Variable	Public Sector and NGOs	For-Profit Private Sector
Variable	Model 15	Model 16
Time (Based on 2014)	0.048 (0.070)	−0.123 *** (0.050)
Implementation of the Policies (Mainland as the Control Group)	−0.101 (0.0809)	−0.294 *** (0.066)
Interactive Term (Time × Implementation of the Policies)	0.039 (0.108)	0.225 *** (0.077)
Cons	1.790 *** (0.319)	2.612 *** (0.225)
Control Variable	Controlled	Controlled
R2	0.096	0.073
F	4.256 ***	6.178 ***
N	515	1061

*** in the table indicate that the standardized regression coefficients reach the significance level of 0.01. Standard errors are in parentheses.

**Table 11 behavsci-13-00644-t011:** Test results of the subsample for social class.

Group	Lower-Class Group	Middle-and Upper-Class Group
Variables	Model 17	Model 18
Time (based on 2014)	−0.142 *** (0.042)	−0.0786 ** (0.039)
Implementation of the Policies (Mainland as the Reference group)	−0.196 *** (0.074)	−0.249 *** (0.048)
Interaction Items (Time × Implementation of the Policies)	0.116 ** (0.090)	0.137 *** (0.059)
Cons	2.738 *** (0.206)	2.821 *** (0.163)
Control Variables	Controlled	Controlled
R2	0.0592	0.0768
F	5.775	10.84
N	1222	1724

*** and ** in the table indicate that the standardized regression coefficients reach the significance level of 0.01 and 0.05, respectively. Standard errors are in parentheses.

## Data Availability

The data for this study can be downloaded from the official website of the World Values Survey: https://www.worldvaluessurvey.org/wvs.jsp (accessed on 1 May 2023).

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
