# Peer review of "How China’s Great Bay Area Policies Affect the National Identity of Hong Kong Youth—A Study of a Quasi-Natural Experiment Based on the Difference-in-Differences Model"

_behavsci, 2023, doi:10.3390/bs13080644_

Round 1

Reviewer 1 Report

Thank you very much for this opportunity to revise the manuscript titled "How China’s Bay Area Policies Affect National Identity of Hong Kong Youth -- From a Study of Quasi-Natural Experiment Based on Difference-in-Differences Model" that was submitted to Behavioral Sciences (Special Issue - Intergroup Relations and Social Cognition: Promoting Social Harmony).

Сongratulations for choosing a topical issue. Some interesting results are obtained and the findings have practical implications. 

I would recommend to the authors to supplement the article with the following sources:

doi:10.1111/jasp.12622

doi:10.3390/socsci6030092

doi:10.3390/soc12060181

doi:10.17759/chp.2022180314

doi:10.3390/soc12040117

It is also, suggestions for future studies should be added, substantially.

Please do have this paper carefully edited by a native speaker to make it readable.

This paper holds actual value to the readers on Behavioral Sciences.

I will be happy to review the revised manuscript.

Please do have this paper carefully edited by a native speaker to make it readable.

Reviewer 2 Report

It is hardly surprising that among those youth most positioned to benefit from the policies enacted the greatest acceptance and change in values/attitudes would occur.  The separating of the groups by whether they are positioned to benefit from programs as a quasi experimental analysis begs the question of random selection that all statistical significance tests are based on.  Still, it is good to get confirmation from data that what we would expect to occur does seem to occur.  More interesting would be any evidence that those not meeting the conditions for participation might develop more negative attitudes and be the basis for a resisteance movement.

Reviewer 3 Report

  • The role of policy in shaping culturally-based identities is an important area of research, especially in readily developing social geographies. Below I offer what are intended to be relatively minor edits that will better contextualize the study for a broader audience.

  • The authors’ state that this paper focuses on wether and how early CBA policies affected HK youths’ national identity (line 103), but does not clearly describe how either increased or decreased identity is a problem or a benefit. Perhaps provide several examples of conflicts in the current socialization process (noted in line 152).

  • In discussing macro- and micro-level influences of social identity, the authors suggest that national identity is in part shaped by responses to micro-level influences (line 142). Yet, there is a bit more to discuss here in how macro-level influences affect micro-level factors that may differ geographically and demographically across space and time. 
  • (Line 194), The authors’ statement that HK media distorted facts about Mainland China would be stronger if also supported by examples. 
  • (Line 221), The authors’ claim entrepreneurship and employment are linked to sense of gain and happiness, as a pathway to national identity - support and cite. 
  • The authors provide several example of Mainland China’s efforts to support socialization of HK youths (line 218 - 233), largely in education HK youth in the differences they may experience in Western and Chinese politics, cultures, and economic domains. The authors’ assumption, repeated throughout the paper, is that the socialization process is one directional. Are there any efforts representative of the opposite direction to socialize Mainland residents to HK cultures?
  • (Line 280), the authors’ note HK youth exposure to “liberalized” media and their “irrational” imagination of Mainland government. Support and cite. Also, that this point in the paper, the authors’ seem to be focused on the role of media communication prior to (and perhaps since) 1997. If critical to the study, perhaps this factor should be included more explicitly. 
  • I am not sure Hypothesis #5 is necessary. It seems to be the theoretical summary of the previous hypotheses, but not uniquely testable on its own. 
  • (Line 447), consider reorganizing Hypothesis 6 as H3a, H3b,…etc. 
  • Table 2: not clear if this included only HK youth group, or all groups.
  • Table 2 (but throughout other results): The co-efficients reported, while significant, are very small. This is not uncommon in large-scale types of tests, where variables likely co-vary and are responsive to many external conditions. But it should be acknowledged by the authors and/or placed in context with other similar studies. The results may be significant, but not practically different, or practically valuable in the context of allied studies. (That same comment could apply to the r-square values, except for Model 4 and Model 8 - the difference might be worth discussing). For example, we see much higher values in Table 7. 
  • Figure 3, path analysis diagram, does not quite make sense as drawn. For example, the interaction term is actually much closer to national identity and mainland government trust than mainland government trust is to national identity. Administrative cultural distance virtually overlaps with mainland government trust compared to national identity, perhaps suggesting covariance (which is not necessarily in disagreement with other results or the author’s hypothesis). This comment may seem a minor graphic adjustment, but it suggests something important in how the results may be interpreted. 
  • The authors’ may consider the above comments in the context of their discussion and conclusions. 

The paper contains several minor editing issues (typically footnote or reference numbers in text). 
